# Self and Place Constructs in Climate Change Vulnerability Assessments: Gaps and Recommendations

## Charles Herrick

DC Center, New York University, Washington, DC 20016, USA; ch133@nyu.edu

**Abstract:** In the United States, climate change vulnerability assessments are usually conceived as objectified exercises, based on theoretical orientations such as rational choice or systems theory. They adopt sectorial or population-level frames of reference and are operationalized by means of aggregating mathematical models, geospatial analytical platforms, and advanced visualization tools. While vulnerability assessments are intended to inform decision making, they often lack process-based mechanisms that enable them to be framed in terms of localized knowledge and perspectives. This is a weakness because occupant attitudes regarding places can spark unyieldingly negative reactions to expert-generated, objectivist vulnerability assessment processes and their outputs. In this paper, I attempt to demonstrate the salience of self and place constructs and explore the implications of their tendency to block serious reflection about the nature of potential vulnerabilities and risk management interventions. If acknowledged and addressed in a manner that is empathetic and context sensitive, it may be possible to channel these perspectives to elevate and deepen dialog about climate change and help to identify and compile circumstantially appropriate menus of adaptation policy interventions.

**Keywords:** sense of place; sense of self; self-identity; climate change; vulnerability assessment; risk characterization; adaptive capacity; coping





## 1. Introduction

Adaptive planning for climate change can lead to policy interventions that involve loss of opportunities, foreclosure of private property, mandatory structural modification, community-scale abandonment, changes to occupations and professional activity, and disruption of group lifeways [1,2]. Issues such as these have been known to prompt reflexive apprehension and even overt hostility to exercises in adaptive planning. Although obstructive reactions are frequently trivialized or cast in a pejorative light, stakeholder reluctance to embrace vulnerability assessment processes and outcomes can be a byproduct of constructions of self and place that are deep-seated and unlikely to change, resulting in or contributing to a sort of "imaginative intangibility" regarding some types of risk and their implication with respect to the characterization climate change vulnerabilities [3–5]. Constructions of self and place can make it easy for some stakeholders to deflect serious discussion of climate-related vulnerabilities and associated policy interventions and even encourage 'gaming' during planning exercises [6]. Often derided as NIMBYism, resident apprehensions to objectified, third-party climate vulnerability assessment is a topic worthy of cross-disciplinary reflection and has led to efforts to revise and augment vulnerability assessment conventions, methodologies, and associated processes. If addressed with sensitivity and a flexible outlook, it is possible to accommodate these types of self-perceptions to sharpen the relevance of vulnerability assessments and inform the design and implementation of circumstantially appropriate adaptation management regimes [7–9].

In the first section of this essay, I summarize anthropological and sociological concepts that deal with individuals and their relationship with specific places, with a focus on how constructions of self and place can become comingled. Drawing on the U.S. experience,

the second section introduces and defines concepts that serve as the building blocks of vulnerability—exposure, sensitivity, and adaptive capacity—and explains the determinative role that they assert in the context of assessment exercises. This section also provides a snapshot of state-of-the-practice vulnerability assessment methodologies, illustrating how self and place constructs are currently underrepresented. Focused on Tangier Island, Virginia, section three is a case illustration of how self and place constructs can impact occupant perceptions of climate-related vulnerabilities and—in turn—influence support for alternative policy interventions. Section four explores two avenues through which constructions of self and place can be better approached, captured, and considered in the context of climate change vulnerability assessment. A final section briefly discusses research needs and other issue areas in which self and place constructs may be germane.

## 2. Constructions of Self and Place and the Cultural Mediation of Climate Change Vulnerabilities

Places are not just coordinates on a map or geospatial phenomena, they are culturally mediated constructs. Culture is defined as "symbols that express meaning, including beliefs, rituals, art and stories that create collective outlooks and behaviors, and from which strategies to respond to problems are devised and implemented." [10]. As defined by Edgar Schein, " . . . culture is the pattern of basic assumptions that a given group has invented, discovered, or developed in learning to cope with its problems of external adaptation and internal integration, and that have worked well enough to be considered valid, and therefore, to be taught to new members as the correct way to perceive, think, and feel in relation to problems." [11]. To paraphrase Ward Goodenough's seminal formulation, culture consists of standards for deciding what is, what can be, how one feels about it, what to do about it, and how to go about doing it.

Places, then, can be a composite of geography, human relationships, and symbolic meaning. It has long been recognized that people ascribe particular qualities to places, including manifestation of cultural values and beliefs [12,13]. To paraphrase Hess and others, places can act as a sort of container for episodes of human history; and in so doing, acquire a status that is decidedly affective rather than merely descriptive [14]. Places, as Burley and colleagues write, are "reflections of ourselves." [15]. As the anthropologist Jake Kosek summarizes this, people's association with places can involve three conceptual orientations: possession of, attachment to, and embodiment within [16]. Constructions of place are not evanescent or transactional, but endure over long periods of time and through a wide range of circumstances, nourishing profound fealty for the ways of life that facilitate particular models of occupancy or modes of survival [14]. Further, "once a particular view of place becomes dominant, a 'perceptual lock in' occurs that is difficult to change . . . " Once established, "a place meaning achieves coherence and longevity . . . [providing] a justification for certain types of events while limiting others and gives places an identity not in terms of an open and changing milieu but as a statement of truth." [17].

The concept of self-identity is related to and sometimes bound up with sense of place [10,14,18]. Self-identity can be associated with or based upon a variety of factors, including geographic domains or features, occupational roles, familial or ancestral lineage, and/or the places where people live, work, worship, and interact [15]. A person's relationship with a place may encompass, structure, or color aspects of their identity, possibly coming to stand for character traits such as forbearance or independence. As described by Goffman, self-identity involves the character traits, roles, values, and attitudes that people demonstrate and articulate to others [19]. The relationship between person and place can assume a deep and elemental role in an individual's life; indeed, a relatively new concept, solastalgia, deals with degradations in people's emotional and mental health due to negative transformations of their home environment [20,21].

The culturally imbued character of places implies that they "come alive" through stories or other narrative-based modes of communication and social interaction. Stories unfold in different ways, resulting in diverse genre: drama, tragedy, humor, memoir, and so forth [22]. However, in all cases, there is some point to the telling of the story, some type

of moral or lesson. It stands to reason, then, that climate-induced changes to a particular geospatial domain will not necessarily translate into a perceived 'vulnerability' because they can be accommodated, possibly even dismissed, within other aspects of a place's story line. This means that the transition between a projected change in a climate-related variable, a resultant geophysical impact, and the situation experienced by an occupant is a highly contingent, interpretive, and dialectic phenomenon.

## 3. A Snapshot of Current, State-of-the-Practice Climate Change Vulnerability Assessment: Conceptual Orientation, Methodology, and Process

Climate change adaptation planning efforts are routinely informed and evaluated through an objectified process of climate change vulnerability assessment [10,23,24]. In the U.S., the National Climate Assessment (NCA) process has invested in developing vulnerability assessment methodologies, analytical tools, and communication and visualization approaches [23]. A central aspect of the NCA process has centered on establishment of terminological conventions for the basic elements of vulnerability, including exposure, sensitivity, and adaptive capacity:

- Exposure: The concept of exposure is used to describe climate-influenced stress factors and typically includes phenomena such as drought or sea level rise.
- Sensitivity: Sensitivity is defined as "the degree to which a system is modified or affected by climate perturbations." [25]. It characterizes a system's susceptibility to respond to climatic variability or change. Definitions of sensitivity vary across sectors and disciplines [26].
- Adaptive Capacity: Together, exposure and sensitivity combine to form the potential impact upon a given sector. Adaptive capacity is a characterization of that sector's ability to reduce impacts through some type of constructive change.

Vulnerability, then, is a function of a system's exposure, sensitivity, and adaptive capacity. Structured around these semantic conventions, vulnerability assessments tend to unfold in a manner that is systematic and sequential, beginning with the compilation of data and calculation of projected future climate variables (e.g., temperature, precipitation), characterization of resultant impact scenarios (sea level rise, extreme weather events, shifts in plant hardiness zones), and the modeled operationalization of impacts to produce probabilistic characterizations of risk to particular categories of natural or societal systems. After risk-based vulnerabilities have been articulated and calculated, assessments may also include narrative or quantitative analysis of the system's capacity to adapt to projected impacts. In some cases, vulnerability assessments include efforts to identify and guide selection of alternative interventions to reduce the impact of projected changes in climatic conditions [23,27].

While some vulnerability assessment activities are predominantly conceptual, state-of-the-practice assessments in the U.S. increasingly rely upon mathematical models, geospatial information systems, and decision analytical tools to characterize risks and potential vulnerabilities. Vulnerability assessments adopt a variety of structural orientations, framing analyses in terms of populations, geographic domains, economic sectors, or some combination of the three. Sector-based vulnerability assessments are exercises in aggregation and focus on categories such as agriculture/forestry, ecosystems, human and social systems, human health and welfare endpoints, land resources, marine resources, and water resources. Vulnerability assessments can be configured to include or inform subsequent analysis of the costs and benefits of alternative approaches to reduce risk and/or enhance adaptive capacity [23,27].

Despite their structural and thematic variation, most vulnerability assessments tend to be undergirded by theoretical orientations such as rational choice and/or systems theory [28]. Further, while the NCA has acknowledged the importance of stakeholder involvement and has recommended that 'civic discovery' be made part of future assessments, the current state of the practice tends to address climate impacts on abstracted systems and/or system components and does not include robust mechanisms or process

provisions through which to integrate phenomena such as self and place constructs [23] A recent technical review exercise co-sponsored by the U.S. Centers for Disease Control and Prevention and Environmental Protection Agency identified and evaluated some two dozen definitions of climate change-related vulnerability and found that none explicitly addressed interactions or potential dialectical relationships between and among physical impacts and place constructs [27].

Climate change modelers project a range of impacts to various natural and socioeconomic systems, some of which are projected to be severe, such as the inundation of barrier and other low-lying island systems [29]. However, it is well demonstrated that physical changes to a place that seem potentially catastrophic to a third-party can be viewed as manageable—maybe trivial—to occupants and other stakeholders with a tendency to focus more on social and symbolic relationships than upon any particular physical characteristic [7,10,30]. The example of Tangier Island provides an illustrative basis from which to explore how constructions of self and place influence occupant outlooks and how such perceptions might be captured and addressed in the context of climate change vulnerability assessment [31–33].

### 4. An Illustration of the Salience of Self and Place Constructs: Tangier Island, Virginia

Approximately 18,000 years ago, the landform now known as Tangier Island was a ridge running along the eastern shoreline of the ancestral Susquehanna River. Isolated from the Delmarva Peninsula by the waters of melting continental icesheets, Tangier is now an island situated in the lower Chesapeake Bay. Since the mid-1800s, the landmass of Tangier Island has been reduced by almost 70 percent because of erosion, land subsidence, and more recently, sea level rise due to global warming. Since 1850, the Island has lost approximately nine acres of land per year, resulting in a present-day area of approximately 790 acres. Today, the highest point of land on Tangier Island stands barely four feet above mean sea level. Under mid-level scenarios of sea level rise, much of the remaining land mass is projected to be lost within the next 50 years; and under a high sea level rise scenario, scientists fear that the island will need to be abandoned in as few as 25 years [34]. The population of Tangier peaked in the 1930s at approximately 1500 and has been diminishing ever since, with the 2010 Census registering only 727 inhabitants. Like the landmass, the population of Tangier Island has shrunk dramatically over the years [35].

Most Tangier residents can trace their lineage to a small group of families who settled the island in the late 1700s. Over ten percent of 'Tangiermen' share the name of Parks, nine percent are Pruitt's, and seven percent are named Crockett. In addition to their common ancestry, Tangier residents share a distinctive dialect that bears some resemblance to early English, devotion to a highly individuated sect of evangelical Methodism, and— perhaps most importantly—an occupational lifestyle known as 'Watermen.' As suggested by the name, the Watermen's lifeways are tied closely to the waters of the Chesapeake Bay. Depending on the season, environmental trends, market dynamics, and their personal proclivities, Watermen may harvest crabs, dredge for oysters, or pursue commercial finfish; they may also hire out to pilot or crew on tugboats on the mainstem of the Chesapeake. Whatever their specific form of labor, their lives are tied to the Bay and its rhythms, and especially to the shallow waters that surround and encroach upon Tangier Island [31,33].

The occupants of Tangier Island share a tangible and inviolable sense of place, nourished by long-standing familial ties. Perhaps due to their unrelenting need to battle the forces of nature, residents have derived an ethic of community self-reliance from their parents and grandparents and feel duty-bound to sustain the way of life passed on to them by their families. As Earl Swift explains, Tangier Island—the place—is profoundly fused with the Watermen's concepts of self and is synonymous with character traits such as self-reliance, demonstrable faith, and an almost cavalier perseverance in the face of forces that cannot be controlled [31]. Although the youth move away to follow opportunities, older residents think of leaving Tangier Island as a breach in the ark of their history, or more fundamentally as a moral failure.

As Cross explains, sense of place is sometimes operationalized in terms of three component concepts: place attachment, place identity, and place dependence [36]. *Place attachment* is a bond between a person and a place that evolves as a relationship between conditions and the characteristics of people. In the case of Tangier Island, this can be seen through inhabitant's ready and willed acceptance of the oft-times harsh weather conditions, austere surroundings, and physical dangers that accompany a life on the water. *Place identity* takes the form of emotional and/or symbolically mediated feelings or meanings ascribed to the place by individuals or the broader community. Among the Watermen of Tangier, this can perhaps be exemplified through the tight conflation of natural events with the perceived will of God. For example, an intense thunderstorm is not viewed merely in terms of rain gauge and anemometer readings, but as a gift or opportunity to demonstrate and reveal one's faith in God. Finally, *place dependence* is defined as a functional quality through which the value of a place is manifested by the satisfaction of material needs [36,37]. In the case of Tangier, place dependence is rooted in the island's proximity to what are some of the richest and most abundant blue crab, oyster, and fishing grounds in the world. The island literally 'provides for' its inhabitants. While deconstruction of the three components of place attachment may be an analytically coherent project, I suggest that in the case of Tangier, the individuated components merge into a sort of perceptual amalgam, which while perhaps difficult to articulate, is nonetheless unabating in its hold over how residents view and present themselves to one another and the outside world. Fiercely independent, the Watermen of Tangier cleave to and nurture a different, more nuanced, view of their community than that suggested through state-of-the-practice vulnerability assessment.

While many outsiders and experts cast this island community as caught in a slow state of geophysical and demographic collapse, resident Watermen see things differently. Indeed, the pull of place has a marked influence on how Tangiermen see and relate to the world, especially when it comes to how they view climate change and its impacts. Although keenly aware that their island home is rapidly being consumed by the waters of the Chesapeake, islanders are steadfast—even aggressive—in the expression of a shared position that the problem is not due to climate change and sea level rise, but rather the result of ordinary erosion driven by wave action, compounded in the minds of a few, by the relentless channeling and burrowing behavior of nutria and other unspecified "marsh creatures [31]. Islanders are skeptical of sea level rise, in the first instance, because of a reflexive and pervasive resistance to the views of 'mainland experts.' Instead, they put their faith in Providence and the commonsense wisdom of their community, and in so doing, tend to reject information and direction that originates beyond the island. One resident likens "the outside world to the waves tearing at the shore–erosive and sneaky, worthy of constant vigilance." However, on a more practical level, Tangermen reject claims and projections of sea level rise because they do not "see it" happening [6,31]. They are on the water nearly every day of their lives and simply have not witnessed any change in the level of the sea.

While the erosion issue is viewed as acute, the selection of acceptable adaptations is limited. The very notion of abandonment and relocation is anathema to most islanders. As life-long resident Carol Moore put it, "I'm staying until I got one foot on the side and one foot in the water." Or as another says, "We are here until he [God] says otherwise." [31]. In the minds of Tangiermen, the island's only acceptable mechanism of salvation is construction of a protective ring of segmented breakwaters and dunes encircling most of the remaining land area. They offer up prayer requests for "the seawall" at church; and bemoan inaction and a multi-decadal succession of U.S. Army Corps of Engineers studies that address the costs and benefits of alternative methods to protect Tangier Island.

## 5. Better Reflecting Self and Place Constructs in the Context of Climate Change Vulnerability Assessment

Enhanced sensitivity to the socio-cultural foundations of place calls for modifications to the vulnerability assessment process along at least two major axes: (1) how self and place constructs impact the ways in which people think about—or frame—climate change

and make sense of its impacts; and (2) how these constructs affect people's view of the effectiveness and legitimacy of alternative adaptive interventions.

These factors are explored below.

Analytical Framing and Sense Making As demonstrated through the case of Tangier Island, self and place constructs can affect how people react to problematical situations [5,10,14]. Alteration to physical features or processes associated with a particular spatial domain might or might not result in a consequent transformation in how that place is perceived by residents or other stakeholders. This problematizes the fact that place constructs may be inconsistent with expert-generated, objectified assessments of climate-related impacts and risks. The discontinuities between expert and lay understandings is a topic that has been widely explored in the literature of risk communication and related areas of social science [38,39]. Efforts to mitigate this dissonance frequently hinge upon outreach and educational interventions. While education may be effective in situations where partisans suffer merely from a simple knowledge deficit, it is unlikely to have a meaningful influence on people with a deeply rooted sense of place [40,41] Indeed, people have been known to disregard or even distort information that accompanies public deliberations, if contemplated interventions "run counter to their sense of place and selves in an environmental . . . setting." [42].

So how can place constructs be addressed in the context of climate change vulnerability assessments? As recognized by Chapin and Knapp, in situations where conceptions of place are sharply contested, vulnerability assessment and adaptive planning exercises are "best fostered by transparent and respectful dialog to identify shared values and concerns and negotiate areas of disagreement" (2015, 38) A vibrant and wide-ranging literature addresses deliberative, inclusive, and co-productive approaches through which to frame and evaluate environmental and resource management planning and policy interventions [10,24,43–46]. Informed by a variety of socio-scientific disciplines, this literature constellation supports the proposition that it is worthwhile to establish a foundation of trusting dialog before trying to engage stakeholders in an effort to characterize climate change vulnerabilities [43,45,47,48]. Building on accumulated social capital, the goal of such exercises would be to co-evolve an enhanced understanding of ways in which impacted parties "think about" climate-induced changes to the places with which they identify [49]. Investment in an effort to coalesce data and analytical outputs around incumbent constructions of self and place provides a sort of platform upon which to articulate characterizations of climate change-related impacts that are, as James Scott says, "legible" to impacted parties [50].

It is critical, then, that objectivist assessments of climate vulnerability do not merely present stakeholders with a series of draft outputs, but rather undertake early dialog in an effort to first, establish trust, and then, frame questions in terms of perspectives that are tangible and recognizable to occupant stakeholders. Efforts to engage and facilitate such dialog can draw from a wide variety of proven tools and scholarly orientations. Insights about particulate instances of place identity can best be captured through qualitative, elucidating processes and measures. As summarized by Lewicka, qualitative measures to characterize sense of place can be roughly divided among verbal and pictorial measures [44].

Proven and commonly utilized verbal measures include in-depth interviews (structured or semi-structured) with subsequent analysis of recorded content [10,30]; workshop-style discussions [45]; think-aloud protocols [51]; reports derived through focus groups and micro focus groups; facilitated responses to selections of statements, with or without participant outputs from participant ranking exercises [52]; reviews of oral history [2]; appreciative inquiry [48]; story telling; and free association exercises [53] Sometimes used in conjunction with verbal measures and discursive analyses, pictorial methods are based on a wide variety of visual queuing and representation mechanisms including photographs, drawings, maps, schematics, objects and artifacts, and multi-media collages. Materials may be prepared by a research-facilitation team or prepared by participants themselves [54,55]. Some commonly used tools include evaluative maps, influence mapping, and depictions of decision trees and planning cycles [48,56]. Once place characterizations and measures have

been captured, there are at least four broad approaches available for dealing with contested concepts and/or divergent constellations of perspective. As described by Chapin and Knapp (2015), these tool sets include discursive analyses, boundary concept theory, incompletely theorized agreement, and common property theory, all of which have been utilized to identify commonalities and differences within discourse and to provide techniques and protocols for conflict resolution.

Activities such as these could be hosted and facilitated by boundary organizations with existing networks that could help to bridge local values with technical inputs [57,58]. Again, the point of such a process would be to facilitate a convergence of perspectives and help to establish the basic legitimacy of climate adaptation as a matter of legitimate community action.

The Perceived Legitimacy of Adaptive Interventions Selection of policy instruments appropriate to address specified resource management objectives has long been a challenge for decision makers and applied policy analysts [59–61]. Alternative instruments are evaluated in terms of considerations such as overall effectiveness, certainty of achieving desired outcomes, balance of benefits and costs, burden on targeted parties, efficiency, and ease of administration. Different configurations of instruments entail different total costs as well as differing distributions of cost among targeted parties. As we have already emphasized, some adaptation interventions involve actions that impact communities and individual citizens in a deeply affective and/or material fashion, such as condemnation and forced abandonment of property, physical alteration of structures, revocation or imposition of easements, and other enforced changes in land or resource use. Policy interventions available to reduce vulnerability and enhance adaptive capacity are diverse, with most instruments being broadly applicable across a range of vulnerability and impact categories [7,62].

When it comes to vulnerability assessments, stakeholder participants are usually very aware that they are not merely part of an 'academic' exercise, but perhaps setting the stage for future changes in public policy. As I have emphasized, self and place constructs can make it hard for some individuals to seriously consider or even understand externally designated vulnerabilities. This, in turn, interferes with their ability to weigh, much less accept, policy interventions based on characterizations of vulnerability that they find illegible. Consider again the example of Tangier Island: while condemnation and abandonment might seem like expedient risk reduction interventions to many—perhaps most—citizens of Virginia, institution of such a policy regime would undoubtedly seem radically dissonant to multi-generational, hereditary residents who see themselves as 'duty bound' to sustain the place that serves as their home, underpins their occupation, and provides the font of their self-identity. Asking these individuals to consider seriously a policy of abandonment would force them to act in a manner that would be perceived as discordant with their moral compass and sense of familial duty. For this reason, it would be advisable to structure stakeholder deliberative processes and menus of vulnerability enhancement options in a manner that acknowledges prevailing occupant constructs. An adaptive planning menu for Tangier might involve:

- Enhanced regulatory oversight of new construction within designated areas, augmented by a state-level enhancement of insurance coverages for existing property owners, perhaps including government subsidization for insurance.
- Building codes that require elevation of structures, combined with a grant program to help address the costs of structural modification.
- Construction of a harbor breakwater, shoreline protections, and a system of dunes and native plantings to protect impacted areas and replace lost land area, with funding provided through general revenues or user fees.

Despite the best of intentions, accommodative interventions such as those outlined above may prove untenable. It may be the case that 'draconian' policy interventions—such as abandonment—become necessary. However, even in cases such as this, inputs from a co-productive research process designed to capture and address prevailing place constructs

can be used to inform a sensitive approach to the governance of climate change adaptation. Forced abandonment might be "softened" [2]. through interventions such as the following:

- 'Abandonment' strategies that preserve and/or recreate elements of Tangier's place identity. If residents must be removed from a particular place, perhaps the removal package could also include mechanisms to maintain community solidarity, such as provisions to relocate schools, church buildings and other iconic community fora. It might also include an exclusive state charter to the fishing and crabbing grounds customarily used by the Islanders.
- Provision of public services to help displaced occupants deal with solastalgia and other emotional and psychological impacts associated with attenuation and ruptures in place identity [20,21].

Legitimate and durable policy regimes are more likely to emerge through the conversational, deliberative, and co-productive interaction of occupants and decision makers than through third-party application of abstract, system-focused assessment methodologies intended to characterize and estimate risks and vulnerabilities [63]. To summarize, vulnerability assessments need to facilitate occupant input on the front end of the process and provide a menu of recognizable and legible outputs. This might include a number of practical augmentations to the current state the practice for climate change vulnerability assessment:

- Assure that scenario development and modelling exercises are based on assumptions, boundary conditions, and thresholds that have been informed and framed in light of stakeholder consideration and dialog and are not merely imposed upon the parties to the assessment process through expert elicitation, literature review, or meta-analysis.
- Augment mathematical, probabilistic, and geospatial approaches with narrative rubrics that delineate legible operationalization of variables and characterization of outcomes.
- Flag or perhaps even preclude adaptive interventions that are fundamentally inconsistent with occupant conceptions of self and place.
- Experiment with ways to address potential occupant coping in a non-judgmental manner and avoid imposition of value-laden characterizations such a 'maladaptation.' Exercise caution and sensitivity if attempting to transform narrative accounts of coping into quantitative constructions such as 'coping range' or similar abstractions [64].
- Format reports, websites, and other communication devices to highlight divergences between objectivist assessment outputs and occupant understandings of place. Do not treat occupant understandings as mere 'exceptions' to an objectivist frame of reference, but rather as primary categories of analysis and organization.

It clearly makes sense for vulnerability assessments to include a menu of interventions that satisfy diverse stakeholders and address needs in a manner that is congruent with place constructions and situations [65–67]. Achievement of this vision, while worthy and important, is by no means a trivial matter.

Identification and characterization of place constructs and their incorporation into adaptation policy regimes will, in many cases, require original, case-by-case research, and a correlative commitment of resources, both time and financial. In addition, consideration of place constructs as part of the vulnerability assessment process will demand access to a broader socio-scientific skill set than presently utilized. Resource limitations will likely mean that such a research and assessment enterprise cannot always be conducted, prompting a need for potentially controversial selection and engagement rules or policy decisions to determine which jurisdictions will be subject to elaborated study and which will have to 'make do' with abstracted, system-level assessments of vulnerability. Also problematic, it will likely be the case that co-productive and action-oriented research into place-based values and outlooks will not be fully reflected in subsequent policy interventions. In other words, high quality discursive interactions may create expectations

that policy initiatives fail to meet, leading to greater alienation than might originally have been the case.

## 6. Research Questions and Potential Applicability to Other Issue Areas

A variety of interesting research topics emerge as we ponder the inclusion of self and place constructs in the context of climate change vulnerability assessment, some practical and methodological, others of a decidedly normative character. For instance, it is far from obvious that all self and place constructs deserve equal treatment in the context of vulnerability assessment and/or selection of alternative policy interventions. An eighth-generation resident and a non-occupant real-estate developer might share an ardent interest in a barrier island, but how should we determine whether they 'count' the same? Should we view some types of self and place constructs as "genuine" while deeming others merely "ancillary?" On another front, can people become imbued with meaningful constructions of place for a location which they have never visited or visited only rarely? Can constructions of place be modified through experiential or educational interventions; and if so, what are the moral implications of policy interventions intended to change people's self and place constructs in the service of climate change adaptation? The perspective presented in this essay suggests that the professional climate adaptation community might need to reconsider 'coping' as an acceptable way of dealing with exposure to climate-related stress factors. At present, it is not difficult to find references to coping as a form of 'maladaptation.' [68]. As discussed here, this can be a highly dismissive and unnecessarily pejorative characterization, likely to stand in the way of productive vulnerability assessment exercises. While self and place-constructs seem pertinent to climate change vulnerability deliberation, there clearly remains much to learn about their potential for routine application, especially in a policy context.

Beyond climate change and disaster response, there are environmental and resource management issue areas that entail policy interventions involving alteration to places and people's activity in the context of specific landscapes, including grazing management, sustainable fisheries, ocean planning, sustainable forestry, and endangered species management planning. All involve thorny questions of how to balance social, cultural, and ecological considerations; and all entail the possibility for perceptual discontinuities based upon culturally imbued differences in apprehension, cognition, and valuation. Moreover, there are public policy issues in domains such as public health and community policing that may also hinge on people's sense of self. Further, there are examples of international development initiatives dealing with water resource management or commodity extraction that either impinge upon or take advantage of occupant perspectives of place and self [4]. Self and place-constructs appear to have a currency within and beyond the context of climate change vulnerability assessment.

**Funding:** This research received no external funding.

**Institutional Review Board Statement:** Not applicable.

**Informed Consent Statement:** This research is informed by review of existing literature and involved no interaction with research subjects.

**Data Availability Statement:** Not applicable.

**Conflicts of Interest:** The author declares no conflict of interact.

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
