# Peer review of "Self and Place Constructs in Climate Change Vulnerability Assessments: Gaps and Recommendations"

_sustainability, doi:10.3390/su13052990_

Round 1

Reviewer 1 Report

Adaptation by vulnerable communities to climate change is a pertinent and relevant issue to address. This paper takes a particularly vulnerable community and suggests there is a need for more sensitive dialogue to help them adapt to a disappearing land mass they call home. The paper, however, could do with more focus as the need for more sensitive dialogue that understands the context of place has already been covered in recent papers. 

Soime examples:

Cvitanovic, C., Howden, M., Colvin, R.M., Norström, A., Meadow, A.M. and Addison, P.F.E. 2019. Maximising the benefits of participatory climate adaptation research by understanding and managing the associated challenges and risks. Environmental Science & Policy, 94, pp.20-31

Chapin, FS and Knapp, CN. 2015. Sense of place: A process for identifying and negotiating potentially contested visions of sustainability. Environmental Science & Policy, 53, pp. 38-46. DOI: 10.1016/j.envsci.2015.04.012

Moser, SC. 2014. Communicating adaptation to climate change: the art and science of public engagement when climate change comes home. WIREs Climate Change 5:337–358. doi: 10.1002/wcc.276

Egoz, S.; Jørgensen, K.; Ruggeri, D.; Primdahl, J.; Kristensen, L.S.; Arler, F.; Angelstam, P.; Christensen, A.A.; Elbakidze, M. 2018 Rural landscape governance and expertise: on landscape agents and democracy. In Defining Landscape Democracy; Edward Elgar Publishing: Cheltenham; pp. 153–164 ISBN 9781786438348.

Sense of place is considered to be made up of three aspects, place attachment, landscape or place identity and place dependence. How do these aspects relate specifically to the inhabitants of Tangier Island would bring some more focus and novelty to the paper. In addition little is said of their beliefs or not in the concept of climate change, particularly as they have been identified as conservative evangelicals.

Suggested articles:

Lewicka 2011: Place attachment: How far have we come in the last 40 years?

Cross 2015: Processes of place attachment: An interactional framework. 

It is accepted that there is a need for more sensitive dialogue with communities but how can this dialogue bridge the gap of acceptable responses to loss of home? How can inhabitants be supported to accept or change? Environmental Melancholia (Lertzman 2015) and Solastalgia (Albrecht 2007 -Solastalgia: The distress caused by environmental change and Galway et al. 2019 -Mapping the Solastalgia Literature: A Scoping Review Study) cover the issues they may face.

Specific points in the paper

Page 2 susrvivial. (Hess et al 2008)

Page 5 in the box labelled Exhibit 1

"Today, the highest point of land on Tangier Island stands barely four feet about mean sea level."

This section in the box also ends abruptly with "Although the youth move away to follow opportunities, older residents..."

Would suggest this box is titled "An example of Self, Place, from Tangier Island, Virginia" or something similar as there are no other exhibits.

In conclusion, more focus on the specific contexts of the case study would help to draw the lessons of where dialogue would be useful and in what way and the addition of more up-to-date references. 

Author Response

Please see attached comment-by-comment tabular summary of responses. Thank you.

Reviewer 2 Report

This Essay reflects on the role of the Climate Vulnerability Assessments (CVA) in informing decision making and most importantly its shortcomings, namely the lack of “process-based mechanisms that enable them to be framed in terms of localized knowledge and perspectives. This is a weakness because occupant attitudes regarding places can spark unyieldingly negative reactions to expert-generated, objectivist vulnerability assessment processes and their outputs.”

In theory, I do agree that this is often happening, reflecting a gap between more nuanced local circumstances and how they are portrayed by researchers who are sometimes even unfamiliar with the place-based factors of locations they are assessing. However, I think researchers working in the domain of CVA are aware of this shortcoming and there is a growing body of literature that is not addressed in this text that calls for the participatory input and more transparent and nuanced approach that will improve policy relevance of CVAs. Also, it may be useful to mention the barriers of doing this and also discuss the inherent role of CVA – which is not to be taken at its face value but rather to indicate potential areas of concern and call for further investigation. I suggest a more balanced approach between strengths and weaknesses of CVA - sometimes this is the only approach we may have available to better understand what is going on on a larger scale. The arguments would seem more balanced if a progress with CVA process would be described especially reflecting on a more recently published work since 2015, e.g.,:

Jurgilevich, A., Räsänen, A., Groundstroem, F., & Juhola, S. (2017). A systematic review of dynamics in climate risk and vulnerability assessments. Environmental Research Letters12(1), 013002.

De Sherbinin, A., Bukvic, A., Rohat, G., Gall, M., McCusker, B., Preston, B., ... & Zhang, S. (2019). Climate vulnerability mapping: A systematic review and future prospects. Wiley Interdisciplinary Reviews: Climate Change10(5), e600.  

Otto, I. M., Reckien, D., Reyer, C. P., Marcus, R., Le Masson, V., Jones, L., ... & Serdeczny, O. (2017). Social vulnerability to climate change: A review of concepts and evidence. Regional environmental change17(6), 1651-1662.

David-Chavez, D. M., & Gavin, M. C. (2018). A global assessment of Indigenous community engagement in climate research. Environmental Research Letters13(12), 123005.

This is a very relevant topic and I do agree that bottom-up approach that integrates attitudes, opinions, and preferences of occupants is very important, especially now that there is an emerging push for managed retreat and similar top-down adaptation interventions that seem to advocate for a cookie-cutter approach to problem solving. Thus, to really enable researchers to integrate self- and place-based determination as essential for the selection of appropriate adaptation pathway, it would be good to discuss some of the barriers such as primary data collection that is often necessary to achieve this, lack of thrust in science/outsiders/government, past negative experiences with researchers/policy makers that may prevent any future interactions with certain subpopulations, limited resources, etc. Some of these aspects are also quite applicable to Tangier Island case study. And then, what can be done to overcome those barriers. Meaning, how to implement it…

Title could be more concise – e.g., Self- and Place-Constructs in the Climate Change Vulnerability Assessments: Gaps and Recommendations

Exhibit 1 – please consider renaming to Case Study 1 – it seems that the text box abruptly ends with “Although the youth move away to follow opportunities, older residents” and is missing additional text. The narrative is nice and emphasizes how circumstances change and affect culture, traditions, and livelihoods of many remote, isolated populations. In its current state, story is compelling but does not clearly indicate how it relates to the CVA – I suggest that authors add some explicit text on this.

OK, I just came across this text – I think it can be added to the textbox or the Case Study content can be added under the regular text to improve the flow and readability.

For sentence: “However, it is well-demonstrated that physical changes to a place that seem potentially catastrophic to a third-party can be viewed as manageable – maybe trivial - to occupants and other stakeholders with a tendency to focus more on social and symbolic relationships than upon any particular physical characteristic.” Please provide citations and clarify “symbolic.”

Also, author/s at times uses “I’ and then “We” – please be consistent.

Author Response

Please see attached comment-by-comment tabular summary. Thank you.

Round 2

Reviewer 1 Report

The author has integrated the suggestions made in the previous review in a thorough and engaging manner. The author has also brought out the characteristics of the inhabitants of Tangier Island that present the challenges to adaptation to climate change in a sensitive and understanding way. 

A very minor detail in the text

"Consider again example of Tangier Island" is missing a "the"

This manuscript is a resubmission of an earlier submission. The following is a list of the peer review reports and author responses from that submission.